# Pressure-Steam Heat Treatment-Enhanced Anti-Mildew Property of Arc-Shaped Bamboo Sheets

**DOI:** 10.3390/polym14173644

**Published:** 2022-09-02

**Authors:** Xingyu Liang, Yan Yao, Xiao Xiao, Xiaorong Liu, Xinzhou Wang, Yanjun Li

**Affiliations:** 1Jiangsu Co-Innovation Center of Efficient Processing and Utilization of Forest Resources, Nanjing Forestry University, Nanjing 210037, China; 2Hangzhou ZhuangYi Furniture Co., Ltd., Hangzhou 311251, China; 3Bamboo Engineering and Technology Research Center, State Forestry and Grassland, Nanjing 210037, China; 4Dareglobal Technologies Group Co., Ltd., Danyang 212310, China

**Keywords:** bamboo, pressure-steam heat treatment, bamboo cell wall, anti-mildew property

## Abstract

Bamboo is one of the most promising biomass materials in the world. However, the poor anti-mildew property and poor dimensional stability limits its outdoor applications. Current scholars focus on the modification of bamboo through heat treatment. Arc-shaped bamboo sheets are new bamboo products for special decoration in daily life. In this paper, we reported pressure-steam heat treatment and explored the effect of pressure-steam on the micro-structure, crystallinity index, anti-mildew, chemical composition, physical properties, and mechanical properties of bamboo via X-ray diffractometer (XRD), scanning electron microscopy (SEM), Fourier-transform infrared (FTIR), wet chemistry method and nanoindentation (NI). Herein, saturated-steam heat treatment was applied for modified moso bamboo for enhancing the anti-mildew properties and mechanical properties of moso bamboo. Results showed that with the introduction of saturated steam, the content of hemicellulose and cellulose decreased, while the lignin-relative content increased significantly. The anti-mildew property of moso bamboo was enhanced due to the decomposition of polysaccharide. Last, the modulus of elasticity and hardness of treated moso bamboo cell walls were enhanced after saturated-steam heat treatment. For example, the MOE of the treated moso bamboo cell wall increased from 12.7 GPa to 15.7 GPa. This heat treatment strategy can enhance the anti-mildew property of moso bamboo and can gain more attention from entrepreneurs and scholars.

## 1. Introduction

Bamboo is an indispensable and ideal material for alleviating the shortage of wood resources in modern society [1]. Bamboos are widely used in the fields of construction, decoration, building and daily necessities [2]. Currently, bamboo dominates the wood-processing industry due to its excellent mechanical property, short growth cycle, and easy harvest [3]. For outdoor application, bamboo-based products such as bamboo scriber, flattened bamboo boards, bamboo flooring boards, and so on, are easily affected by fungi, UV, and water due to the abundance of polysaccharide and poor dimensional stability [4]. Thus, the poor anti-mildew property and poor dimensional stability limits their outdoor applications. Shortage of woody materials, enhancing utilization of bamboo materials, and the move towards an enhancement of dimensional stability and mechanical properties of moso bamboo have prompted the development of bamboo modification [5]. Thus, heat treatment has attracted more attention from scholars and scientists due to its eco-friendliness and cost-effectiveness in comparison to chemical modification and other modification methods. In addition, the research on the biodegradability of bamboo or woody materials is a research gap in current research [6].

As we known, bamboo culms consist of hemicellulose, cellulose, lignin, ash, and so on [7,8,9]. The main chemical composition mentioned above contains amorphous phases. When exposed to high-temperature and high-pressure conditions, these chemical components exhibit viscoelastic and plastic behaviors. Until now, chemical agent impregnation, chemical modification, and thermal modification can effectively enhance the mechanical properties and anti-ant and anti-mildew properties of bamboo samples. Unfortunately, the utilization of chemical agents is not environment-friendly and can be harmful to the human body when these bamboo-based products are applied in our daily life. Over the years, dry steam, inert gas (nitrogen), and water were usually used as a heat treatment medium with a treatment temperature between 150 °C and 230 °C and a treatment time of 1–6 h. The heat treatment reduces the hygroscopic property of the bamboo, consequently reducing the hygroscopic property of the bamboo, thus reducing its shrinkage and swelling properties and improving the dimensional stability when the treatment is above 150 °C. At 180 °C or higher, heat treatment can significantly improve the anti-fungi property of bamboo. However, high temperature also leads to a decrease in its mechanical properties and also to the decrement of moisture content in bamboo [10]. Fortunately, pressure-steam is an effective and environment-friendly heat treatment medium in bamboo processing factories. Pressure-steam is the steam that is in equilibrium with heated liquid water at the same pressure, which has not been heated more than the boiling point for that pressure. In addition, pressure-steam can provide pressure, high temperature, and high moisture content in sealed equipment that can modified bamboo quickly. Bamboo is an anisotropic biomass material, which consists of many cells that are oriented in the axial and radial directions. Therefore, knowledge about the specific molecular mechanical phenomena at the cellular and subcellular levels is of great importance for understanding the effects of thermal modification. Additionally, the visco-elasticity of bamboo restricts its application in large structures that require long-term loading. Thus, it is of great interest to investigate the hardness and modulus of the elasticity of bamboo cell walls as a function of the parameters of thermal treatment. Traditional work focused on the effects of heat treatment medium on the macro-mechanical properties such as shrinkage ratio, bending strength, density, equilibrium moisture content (emc), and so on. Although they performed excellent work on the bamboo thermal modification, they did not explore the micro-mechanical properties of bamboo cell walls. Nanoindentation (NI) is a useful technology that can analyze the nano-mechanics of bamboo from the cell-wall level. For analyzing the effects of pressure-steam heat treatment on the nano-mechanics of bamboo samples, NI was used. In addition, for outdoor application, the anti-mildew properties of treated bamboo samples are also important [11,12,13]. Thus, the anti-mildew properties of the untreated bamboo and treated bamboo samples were also investigated in this paper. Thus, the effect of pressure-steam on the micro-mechanical and anti-mildew properties of moso bamboo is still not clear.

In this paper, 6-year-old bamboo was thermally modified with pressure-steam under different temperatures for the same duration. Additionally, we reported pressure-steam heat treatment and explored the effect of pressure-steam on the micro-structure, crystallinity index, anti-mildew, chemical composition, physical properties, and mechanical properties of bamboo via X-ray diffractometer (XRD), scanning electron microscopy (SEM), Fourier-transform infrared (FTIR), wet chemistry method and nanoindentation (NI). In addition, we tested the anti-mildew property of the control and pressure-steam-treated bamboo samples in this manuscript.

## 2. Materials and Methods

### 2.1. Materials and Pressure-Steam Heat Treatment

Six-year-old moso bamboo (Phyllostachys heterocycla) was collected from Gaoan city, Jiangxi, China. The bamboo specimens were collected from the upper layer. The initial moisture content of the specimens was 90%.

Bamboo specimens with dimensions of 1050 × 850 × 10 mm^3^ (length × width × thickness) were prepared for pressure-steam treatment. Then, the arc-shaped bamboo sheets were transferred into the pressure-steam equipment (12R3426-1, Hangzhou Rongda Boiler Container Co., Ltd., Hangzhou, China) for saturated-steam heat treatment at different temperatures for the same duration. Sample A presented the untreated bamboo sample in this manuscript. Sample B and sample C presented the treated bamboo at 160 °C and 180 °C and same duration (12 min), respectively. The heat treatment process of bamboo are shown in Figure 1A–D.

### 2.2. Chemical Components

The change in chemical components (hemicellulose, cellulose, and lignin) were determined by the wet chemistry method. The bamboo specimens were ground into powder with an average dimension of 30–60 mesh. NREL’LAPS were applied for determining the change in chemical components [14].

### 2.3. Scanning Electron Microscopy

Natural bamboo and pressure-steam-treated bamboo specimens with average dimensions of 5 × 5 × 1 mm^3^ (length × width × thickness) were polished for the SEM (FEI Quanta 200, Eindhoven, Holland) observation. The micro-structure of the control and pressure-steam-treated bamboo was placed in a vacuum environment and then observed by scanning electron microscopy.

### 2.4. Measurement of Cellulose Crystallinity Degree

The crystallinity index of the control and pressure-steam-treated bamboo samples was determined by X-ray diffractometer (Ultima IV, Tokyo, Japan) with a rate of 2°/min ranging from 5° to 40°. The crystallinity degree can be calculated as below [15,16,17]:CrI = (I_002_ − I_am_)/I_002_ × 100%(1)
where CrI represents the crystallinity index, I_am_ represents the minimum intensity of the amorphous, and I_002_ represents the maximum intensity of the diffraction.

### 2.5. FTIR

The natural bamboo and softened treated bamboo were powdered into 100–200 mesh size and dried under vacuum at 80 °C for 12 h. The powder from different samples was used for Fourier-transform infrared (FTIR) spectroscopy analysis by Nicolet iS50 FT-IR spectrometer (Thermo Fisher Scientific, Waltham, MA, USA). Data in the wave number range of 4000 cm^−1^–500 cm^−1^ were collected in ATR mode with 64 scans and a resolution of 4 cm^−1^.

### 2.6. EMC, Density, and Bending Strength of Bamboo Samples after Pressure-Steam Heat Treatment

The equilibrium moisture content, bending strength, and oven-density were tested according to GB/T 15780-1995 “Experimental method for physical and mechanical properties of bamboo”. In detail, the bamboo strips with average dimensions of 160 × 10 × 8 mm^3^ (longitudinal × tangential × radial) were prepared by splitting, cross-cutting, and outer-inner layer removing for mechanical testing. Twelve repeated samples of each experimental set were tested in the 3-point bending test. In the density test, the bamboo samples with average sizes of 10 (length) × 10 (width) × t mm (bamboo wall thickness) were prepared from bamboo culms for testing the density by oven-drying method.

### 2.7. Nano-Indentation Method

The instrument used for the experiments in this paper is the G200 Nano-indenter from Agilent, Santa Clara, CA, USA. The instrument combines the nanoindentation head and in-situ scanning imaging functions to achieve precise positioning of indentations at the nanoscale, improving the reliability and accuracy of experiments and thus enabling fine characterization of the mechanical properties of bamboo cell walls.

The experiments were carried out using the XP system quasi-static continuous stiffness testing (CSM) technique with a Berkovich trigonal diamond indenter (Micro Star Inc., League City, TX, USA). The experimental parameters were set to 100 N/m for contact stiffness, 0.5 nm/s for thermal drift, 0.05 s^−1^ for constant strain rate loading, and 45 Hz for simple harmonic force. The test location of bamboo cell walls and typical Nano-indentation load-depth curves are shown in Figure 2A,B.

During the nanoindentation test, the indenter is first pressed into the material and then held when the set maximum load is reached. During the holding period, the material at the bottom of the indenter begins to creep and deform, which is reflected in the fact that the depth of indentation continues to increase over time during the holding period.
(2)$$H=\frac{P_{max}}{A}$$
where *P_max_* is the peak load, and *A* is that the projected contact space of the indents at peak load. The hardness of different treated bamboo specimens can be calculated as following:(3)$$Er=\frac{\sqrt{\pi }}{2\beta }\frac{S}{\sqrt{A}}$$
where *Er* is the combined elastic modulus of both the sample and indenter; *S* is initial unloading stiffness; and *β* is a correction factor correlated to indenter geometry (*β* = 1.034).

### 2.8. Analytical Procedures

The differences between the control and treated bamboo samples were determined by statistical software SPSS (25.0, IBM, Washington, USA) through Tukey’s tests. Additionally, 8 replicates were used to calculate the means for chemical compositions, crystallinity index, modulus of rupture, modulus of elasticity, and hardness in bamboo specimens.

## 3. Results and Discussion

### 3.1. Micro-Morphology of the Control and Pressure-Steam Treated Bamboo Samples

The SEM images of all bamboo samples are shown in Figure 3. The surface morphology of the control was smooth. With the increase of pressure-steam treatment temperature, the lumen volumes of bamboo are gradually distorted and deformed during the treatment. It can be attributable to the high temperature and high pressure [18]. In other words, the high-pressure condition led to the decomposition of chemical composition in bamboo cell walls [19,20,21].

### 3.2. XRD, FTIR, Chemical Composition Analysis

As shown in Figure 4A. To verify the degradation effect of high-temperature saturated steam on the cellular material of the bamboo material, the chemical composition of the bamboo process fibers before and after the treatment was determined by the wet chemical method. As can be seen from Figure 4, the main chemical composition of the untreated bamboo material was 41.3% (cellulose), 21.6% (hemicellulose), and 21.2% (lignin). The cellulose and hemicellulose content decreased by 2.5% and 42.5%, respectively, while the lignin content increased by 4.9%. Hemicellulose is a polysaccharide substance with a low degree of polymerization, relatively poor thermal and chemical stability, and is prone to decomposition reactions such as deacetylation of polysaccharide substances under the action of high temperature and high humidity [15]. In the chemical composition analysis, it was found that at high-temperature saturated-steam vapor, the molecular chains in the non-crystalline region of cellulose are prone to breakage, exposing more. The chemical composition of the cellulose is analyzed and found to be. The hydroxyl groups on the short molecular chains undergo a “bridging reaction” to form ether bonds, which rearrange and crystallize. The relative crystallinity of the treated bamboo process fibers is increased [18]. The results from FTIR confirmed the decomposition of hemicellulose.

In order to further investigate the effect of hydrothermal modification on the functional groups of bamboo fiber, FTIR was applied for this purpose. As shown in Figure 4B, 1238 cm^−1^ is the C-O strength vibrations peak, while the 1731 cm^−1^ is attributed to the C-O and C=O stretching vibrations of hemicelluloses. The intensity of peaks at 1238 cm^−1^ and 1731 cm^−1^ decreased. They are the absorption peak of hemicellulose; the decrement of intensity of peaks at 1238 cm^−1^ and 1731 cm^−1^ represented the decomposition of hemicellulose. During the hydrothermal modification, the cleavage of acetyl groups to acetic acid occurs, which promotes the degradation of hemicellulose further during the thermal treatment. The peak at 1590 cm^−1^ and 1328 cm^−1^ assigned to the aromatic skeletal vibrations and C=O stretching of lignin and the C=O groups linked to the aromatic skeleton barely changed in this study and decreased in the spectra of thermal-treated bamboo with the increasing temperature, which may result from the possible condensation reactions of lignin during the thermal treatment. The intensity of the absorption peak at 898 cm^−1^ decreased significantly. This may be attributed to a small degradation of the “amorphous region” in cellulose due to the high temperature and high pressure [22].

As is visible in Figure 4C,D, the crystallinity was calculated by Segal’s formula. Obviously, treatment parameters contribute positively to the crystallinity index. The higher the treatment temperature, the higher the cellulose crystallinity degree. The mean crystallinity index (CrI) values of the control were 42.7%. Hydrothermal treatment at 160 °C for 12 min has a higher CrI than the control. For example, the crystallinity index increased from 42.7% to 52.3%. The increment of CrI can be attributed to the cellulose becoming more crystalline [23,24,25]. Hemicelluloses have a character of amorphous nature, so the hemicellulose can be more easily hydrolyzed than lignin and cellulose. Additionally, the decomposition of the amorphous parts in cellulose maybe also contribute to this phenomenon. Statistical analysis demonstrated that there is a significant difference between the crystallinity indexes of bamboo samples treated over 160 °C and untreated bamboo samples [26,27,28,29].

### 3.3. Macro-Physical Property of Untreated and Pressure-Steam Treated Bamboo Samples

Figure 5A–D shows the EMC, density, and bending strength of bamboo after pressure-steam heat treatment. As shown in Figure 5A, the EMC of treated bamboo decreased with the increment of treatment temperature. This is because the hygroscopicity of bamboo samples decreased [30]. In addition, the decomposition of the hemicellulose in bamboo cell walls can also contribute to this conclusion. Normally, many free hydroxyl groups exist in hemicellulose. Due to the high-pressure and high-temperature treatment, free hydroxyl groups are greatly decreased. Thus, the EMC of treated bamboo samples decreased. The density of treated bamboo samples decreased with increasing treatment temperature. This can be attributable to the degradation of chemical components in bamboo samples.

The macro-/micro-mechanical properties of untreated and pressure-steam-treated bamboo specimens were presented in Figure 5C,D. As shown in Figure 5C, the initial modulus of rupture (MOR) of the control and pressure-steam-treated bamboo specimens were 8700, 8550, and 7200 MPa, respectively. After treatment by pressure-steam, the MOR decreased by 5.8%. With the increment of treatment temperature, the MOR reduced by 16.7%. According to previous literature, firstly, bamboo is softened under the high-pressure and high-temperature condition and then pyrolyzed during the thermal modification process. When the temperature reaches 180 °C, the hemicellulose decreased significantly due to the pressure-steam heat treatment [31]. Pressure-steam heat treatment significantly decreased the modulus of rupture of the bamboo samples.

### 3.4. Nano-Mechanics of Untreated and Pressure-Steam Treated Bamboo Sample Cell Walls

Nanoindentation (NI) is a useful technology that can analyze the nano-mechanics of bamboo from the cell-wall level. For analyzing the effects of pressure-steam heat treatment on the nano-mechanics of bamboo samples, NI was used to this objective. In Figure 6, we can find that the hardness and modulus of elasticity of the control were 0.52 GPa and 12.7 GPa, respectively. Through ANOVA statistical analysis, we can find that the MOE and hardness of pressure-steam-treated bamboo increased significantly. This is due to the enhanced content of relative crystallinity index and lignin. In addition, according to previous studies, the bending strength and modulus of rupture of bamboo specimens were obtained from the traditional stress–strain relationship. The macro-mechanical properties of bamboo samples showed a decreasing tendency after saturated-steam heat treatment, while the MOE and hardness of bamboo cell walls increased. Normally, the decrement of bending strength is due to the degradation of chemical composition in bamboo cell walls. The micro-mechanical properties of bamboo cell walls are affected for many reasons, such as cellulose crystallinity, lignin content, moisture content, and so on [32,33,34]. Thus, the mechanical properties of the bamboo cell wall play a subordinate role to macro-mechanical properties. For detail, the Young’s modulus values of bamboo obtained from the macroscopic stress–strain relationship increased under elevated compression markedly. In contrast, the mechanical properties of the bamboo cell walls, as expressed by modulus of elasticity and hardness, have shown increasing tendency. The decrement of bamboo density and bamboo’s macro-mechanical properties are due to the decomposition of chemical composition, and the micro-mechanical properties of the bamboo cell walls play a subordinate role in this regard [35,36,37,38].

### 3.5. Anti-Mildew Property of Untreated and Pressure-Steam Treated Bamboo Samples

The anti-mildew property of untreated and pressure-steam-treated bamboo samples were shown in Figure 7. As presented in Figure 7, the initial infection ratios of sample A, B, and C were all 0%. The infection ratio of untreated bamboo samples increased from 0% to 100% in the first 10 days, illustrating that the untreated bamboo has poor anti-mildew properties. However, sample B exhibited a better anti-mildew property than that of the control. Unfortunately, after 16 days, the infection ratio of the sample also reached 100%. It is obvious that the infection ratio of sample C increased slowly and kept a 0% infection ratio in the first 8 days. Sample C exhibited the best anti-mildew property in comparison to those of sample A and sample B. It can be attributed to the degradation of hemicellulose and cellulose in bamboo cell walls. The decrement of polysaccharide and starch in bamboo made a positive contribution to the anti-mildew property of bamboo samples [31,39,40,41]. In addition, the enhanced lignin content in the bamboo surface can inhibit the adhesion between Aspergilus niger and bamboo samples. Thus, pressure-steam treatment can positively enhance the anti-mildew property of bamboo samples. In addition, the summary of the anti-mildew property of treated bamboo in different references are shown in Table 1.

## 4. Conclusions and Future Perspectives

Pressure-steam heat treatment, as a newer bamboo thermal modification method, can effectively improve the anti-mildew property of bamboo materials. In this work, we assess the effect of pressure-steam heat treatment on arc-shaped bamboo sheets through analyzing the change in micro-structure, functional groups, chemical components, and so on. Results showed that with the introduction of saturated steam, the content of hemicellulose and cellulose decreased, while the lignin relative content increased significantly. The anti-mildew property of moso bamboo enhanced due to the decomposition of polysaccharide. Last, the modulus of elasticity and hardness of treated moso bamboo cell walls were enhanced after saturated-steam heat treatment. For example, the MOE of treated moso bamboo cell walls increased from 12.7 GPa to 15.7 GPa. This heat-treatment strategy can enhance the anti-mildew property of moso bamboo and can gain more attention from entrepreneurs and scholars. This work introduces pressure-steam heat treatment to entrepreneurs and scholars and analyzes the thermal mechanism for the thermally modified bamboo samples, which can gain more attention from bamboo factories and researchers in the bamboo industry.

## Figures and Tables

**Figure 1 polymers-14-03644-f001:**
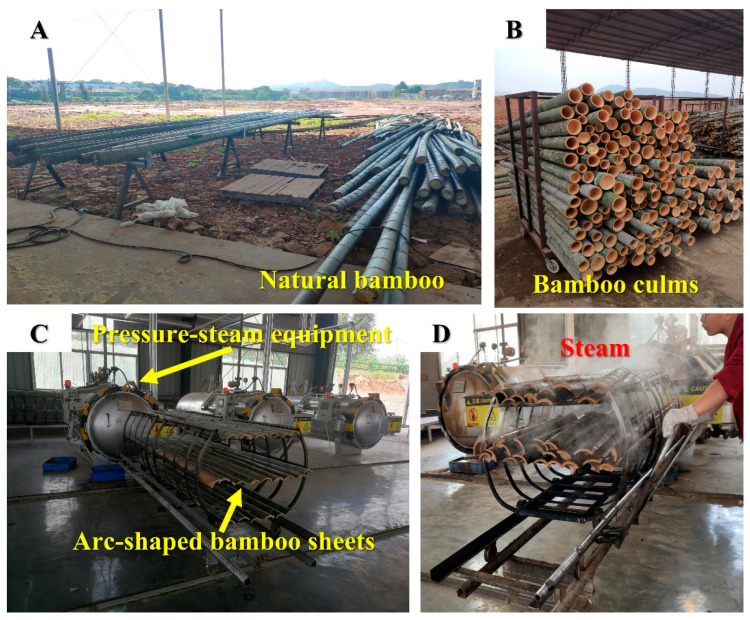
Heat treatment process of bamboo: (**A**) Natural Bamboo in factory; (**B**) Bamboo culms; (**C**) Pressure-steam equipment and arc-shaped bamboo sheets; (**D**) the bamboo sample after saturated-steam heat treatment.

**Figure 2 polymers-14-03644-f002:**
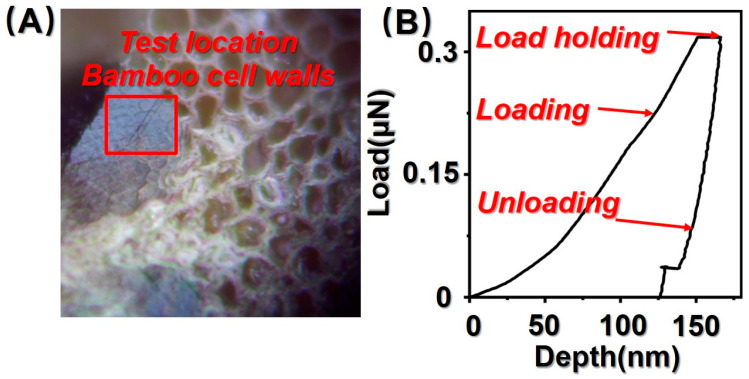
(**A**) Test location of bamboo cell walls; (**B**) typical Nano-indentation (NI) load-depth curves.

**Figure 3 polymers-14-03644-f003:**
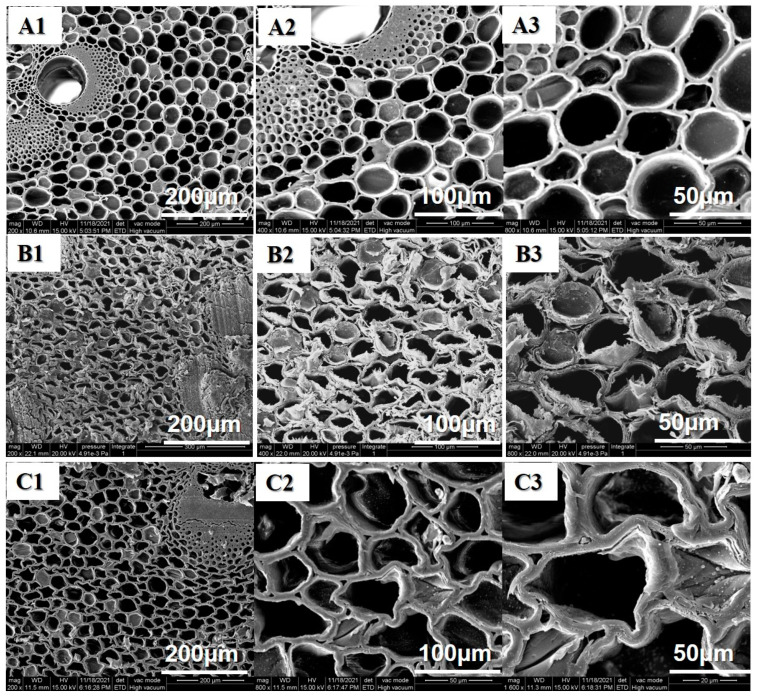
SEM images of different bamboo samples: (**A1**–**A3**) Untreated bamboo sample; (**B1**–**B3**) Softened treated bamboo samples (160 °C/12 min); (**C1**–**C3**) Softened treated bamboo board at 180 °C/12 min.

**Figure 4 polymers-14-03644-f004:**
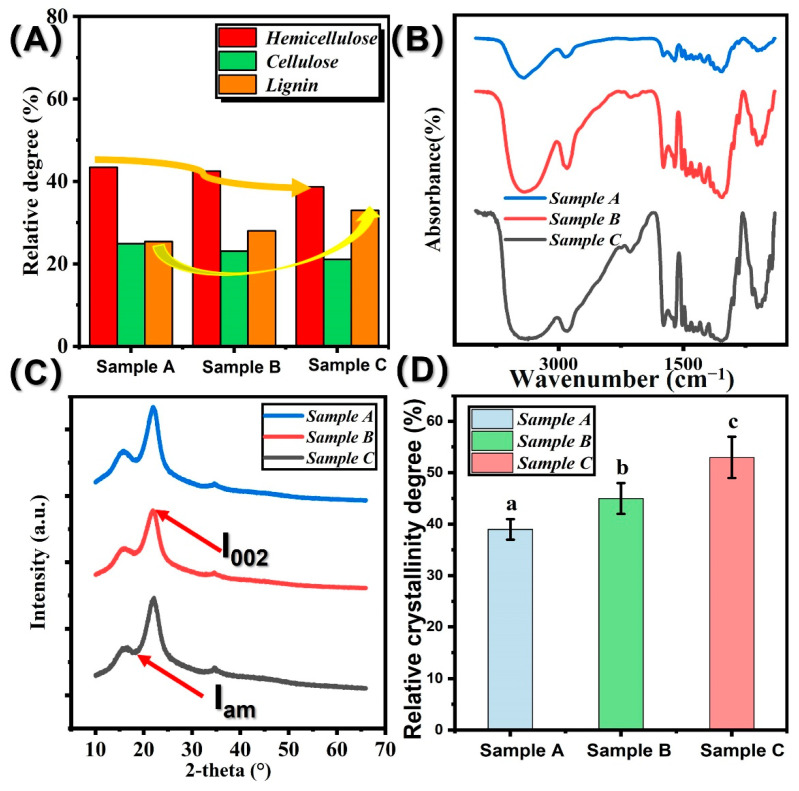
The change in chemical composition, cellulose crystallinity, chemical groups of different bamboo samples: (**A**) hemicellulose, cellulose, and lignin; (**B**): XRD curves; (**C**) Relatively crystallinity degree; (**D**) FTIR curves of different bamboo samples. Different small letters represent the significant difference between heat treatment groups (*p* < 0.05). The error bar in the picture represents the standard deviation.

**Figure 5 polymers-14-03644-f005:**
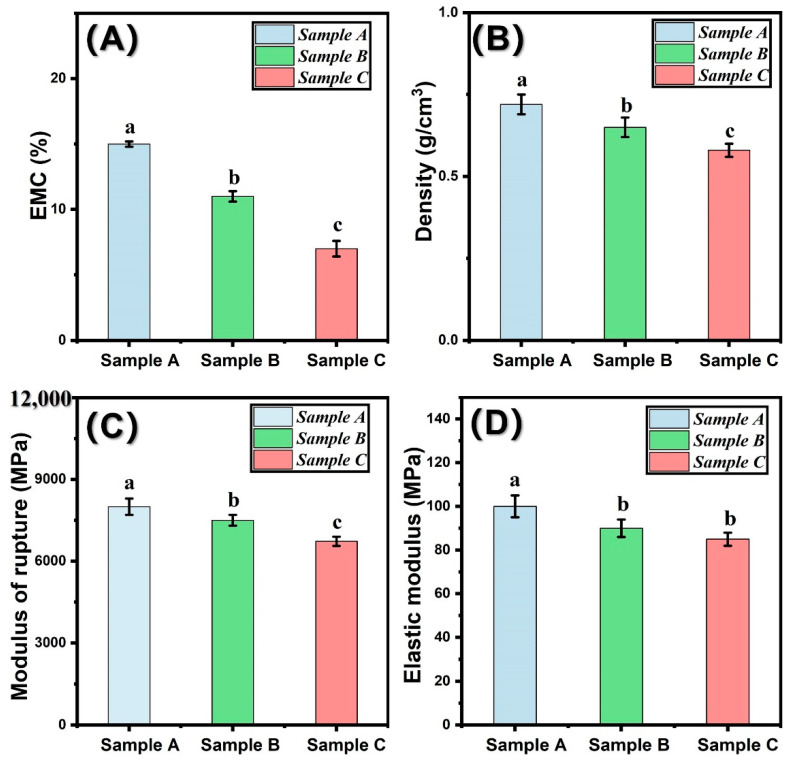
Physical properties of untreated, pressure-steam treated bamboo samples: (**A**) EMC; (**B**) Density; (**C**) modulus of rupture; (**D**) Elastic modulus. Significant difference between heat treatment groups (*p* < 0.05) were presented by different small letters. The error bar in the picture represents the standard deviation.

**Figure 6 polymers-14-03644-f006:**
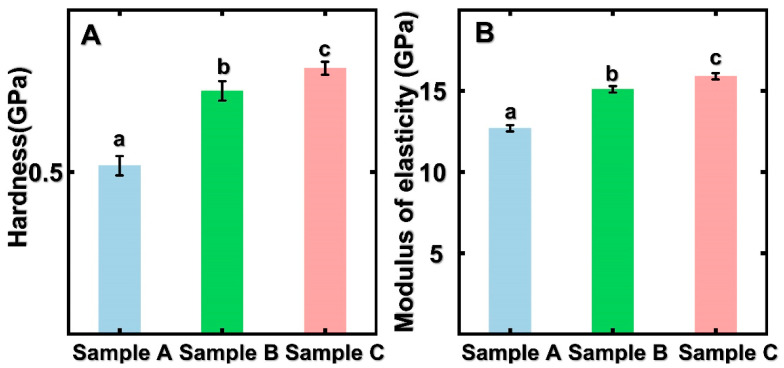
Micro-mechanics of control and treated bamboo samples: (**A**) hardness; (**B**) modulus of elasticity; significant differences between heat treatment groups (*p* < 0.05) were presented by different small letters. The error bar in the picture represents the standard deviation.

**Figure 7 polymers-14-03644-f007:**
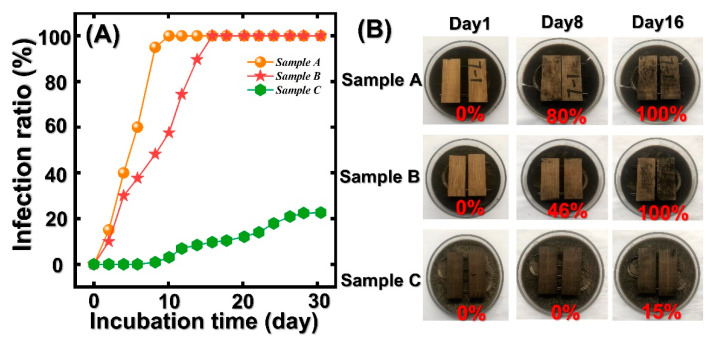
The anti-mildew property of the control and treated bamboo samples in 30 days: (**A**) Infection ratio of untreated bamboo and treated bamboo in 30 days and (**B**) corresponded figures of anti-mildew test.

**Table 1 polymers-14-03644-t001:** Summary of the anti-mildew property of treated bamboo in different references.

Sample	Treatment Medium	Anti-Mildew Properties in 30 Days	Reference
Arc-shaped Bamboo sheet	Pressure-steam	20%	This work
Bamboo	Linseed oil	25%	[23]
Bamboo	TiO_2_	50%	[24]
Bamboo scriber	Fe_3_O_4_	78.6%	[22]

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
