# Peer review of "Pressure-Steam Heat Treatment-Enhanced Anti-Mildew Property of Arc-Shaped Bamboo Sheets"

_polymers, 2022, doi:10.3390/polym14173644_

Round 1

Reviewer 1 Report

1) Is steam treatment of bamboo only acting on surface or 

Author Response

Reviewer 1:

Comments (manuscript ID: polymers-1845589)

Point 1. Authors need to emphasize more on the motivation behind this study and its importance

in an introduction section. Please also include more relevant references.

Response: It is really a detail which should not be neglected as Reviewer suggested and a reference. We have added some motivation behind this study in the introduction, and added some relevant references in introduction section. We are very grateful to Reviewer for reviewing the paper so carefully. We have tried our best to improve the manuscript and have modified some confusing sentences, making them concise and easy to read.

Point 2. Please include the same magnification SEM images for comparison. Soften treated

samples in Fig 3 are not on same scale as other two samples.

Response: It is really a detail which should not be neglected as Reviewer suggested and a reference. We have revised the SEM images in our revised version. We are very grateful to Reviewer for reviewing the paper so carefully.

Point 3. Is the steam treatment only acting on surface or also inside the bulk of the bamboo?

Response: Thanks again to the reviewer on suggesting to properly address the significance of the work. We think the steam treatment both acting on surface and inside the bulk of the bamboo.

Point 4. What is difference between the elastic modulus (Fig. 5) and modulus of elasticity

(Fig.6)?

Response: Thanks again to the reviewer on suggesting to properly address the significance of the work. In fig.5, the elastic modulus is the macro-mechanical properties of bamboo samples obtained by three-point bending test. In fig.6, the modulus of elasticity is the micro-mechanical properties of bamboo cell walls.

Point 5. What is significance of micromechanical properties characterization of bamboo

samples? Why is there increase in micro mechanical elastic modulus with steam

treatment temperature while the macro mechanical characterization showed reverse

trend?

Response: Thanks again to the reviewer on suggesting to properly address the significance of the work. Nanoindentation (NI) is an useful technology that can analyze the nano-mechanics of bamboo from cell wall level. For analyze the effects of pressure-steam heat treatment on the nano-mechanics of bamboo samples, NI was used to this objective.The macro-mechanical properties of bamboo samples obtained from the macro-scale stress-strain relationship. In our manuscript, the mechanical properties of bamboo samples decreased, while the mechanical properties of bamboo cell walls increased. The decrement of bamboo MOR and MOE is due to the decomposition of chemical properties. The micro-mechanical properties of bamboo cell wall were affected by cellulose crystallinity, lignin content, moisture content, and so on. Thus. The mechanical properties of the bamboo cell wall play a subordinate role to the macro-mechanical properties.

Point 6. Fig.7 sample C images are not very clear. Please include good quality images.

Response: Due to the high pressure-steam heat treatment, the color of bamboo samples became darker so that the sample C images shows not very clear, we are apologized for this question. We are very grateful to Reviewer for reviewing the paper so carefully.

Point 7. Define the acronyms wherever used in manuscript

Response: It is really a detail which should not be neglected as Reviewer suggested and a reference. We have defined the acronyms wherever used in our revised manuscript. We are very grateful to Reviewer for reviewing the paper so carefully.

Point 8. Please correct grammatical errors throughout the manuscript

Response: It is really a detail which should not be neglected as Reviewer suggested and a reference. We have corrected the grammatical errors throughout the manuscript. We are very grateful to Reviewer for reviewing the paper so carefully.

Reviewer 2 Report

Greetings, Editor thank you for providing me with the opportunity to review the article. I reviewed the article with title ``Pressued-steam heat treatment enhanced anti-mildew property of Moso bamboo``.  The article topic is intriguing and promising in the area. Overall, the article structure and content are suitable for the POLYMERS journal. I am pleased to send you major level comments, there are some serious flaws which need to be corrected before publication. Please consider these suggestions as listed below.  

1.      The title seems good, but the abstract seems to be wired. Please add one more introductory line of your objective in beginning of abstract.

2.      Research gap should be delivered on more clear way with directed necessity for the future research work.

3.      Introduction section must be written on more quality way, i.e., more up-to-date references addressed.

4.      The novelty of the work must be clearly addressed and discussed, compare previous research with existing research findings and highlight novelty.  

5.      What is the main challenge? Why author choose this material? Please highlight in the introduction part.

6.      Please check the abbreviations of words throughout the article. All should be consistent.

7.      Please don’t use lumpy reference (such as: [1-4,5-9 etc.]). Each reference needs to be properly addressed. Please revise your paper accordingly since same issue occurs on several spots in the paper

8.      The main objective of the work must be written on the more clear and more concise way at the end of introduction section. 

9.      Please include all chemical/instrumentation brand name and other important specification.

10.   Please add chemical reagents section and stated all chemical with brand specifications.

11.   Please provide space between number and units. Please revise your paper accordingly since some issue occurs on several spots in the paper. 

12.   Regarding the replications, authors confirmed that replications of experiment were carried out. However, these results are not shown in the manuscript, how many replicated were carried out by experiment? Results seem to be related to a unique experiment. Please, clarify whether the results of this document are from a single experiment or from an average resulting from replications. If replicated were carried out, the use of average data is required as well as the standard deviation in the results and figures shown throughout the manuscript. In case of showing only one replicate explain why only one is shown and include the standard deviations.

13.   Please provide high quality image for figure 2a.

14.   Each section of the findings requires thorough discussion; a basic explanation is insufficient; please describe each section in a critical way. 

15.   Please add a comparative profile section to compare your results and prove how it better than previous.

16.   Section 4 should be renamed by Conclusion and Future perspectives. Conclusion section is missing some perspective related to the future research work, quantify main research findings, highlight relevance of the work with respect to the field aspect.

17.   To avoid grammar and linguistic mistakes, Major level English language should be thoroughly checked. Please revise your paper accordingly since several language issue occurs on several spots in the paper.

18.   Reference formatting need carefully revision. All must be consistent in one formate. Please follow the journal guidelines. Reference should not in bold form.

Author Response

Reviewer 2:

Greetings, Editor thank you for providing me with the opportunity to review the article. I reviewed the article with title ``Pressued-steam heat treatment enhanced anti-mildew property of Moso bamboo``.  The article topic is intriguing and promising in the area. Overall, the article structure and content are suitable for the POLYMERS journal. I am pleased to send you major level comments, there are some serious flaws which need to be corrected before publication. Please consider these suggestions as listed below.  

Point 1. The title seems good, but the abstract seems to be wired. Please add one more introductory line of your objective in beginning of abstract.

Response: It is really a detail which should not be neglected as Reviewer suggested and a reference. We have added some more introductory line of our objective in beginning of abstract.

Point 2. Research gap should be delivered on more clear way with directed necessity for the future research work.

Response: We are very grateful to Reviewer for reviewing the paper so carefully. We have tried our best to improve the manuscript and have modified some confusing sentences, making them concise and easy to read. We have increase the length of the introduction and explain the novelty of the work. We are very grateful to Reviewer for reviewing the paper so carefully.

Point 3.  Introduction section must be written on more quality way, i.e., more up-to-date references addressed.

Response: We are very grateful to Reviewer for reviewing the paper so carefully. We have tried our best to improve the manuscript and have modified some confusing sentences, making them concise and easy to read. We have increase the length of the introduction and explain the novelty of the work. We are very grateful to Reviewer for reviewing the paper so carefully. We have added more references in our introduction section

Point 4.  The novelty of the work must be clearly addressed and discussed, compare previous research with existing research findings and highlight novelty.  

Response: We are very grateful to Reviewer for reviewing the paper so carefully. We have tried our best to improve the manuscript and have modified some confusing sentences, making them concise and easy to read. We have increase the length of the introduction and explain the novelty of the work. We are very grateful to Reviewer for reviewing the paper so carefully. We have added more references and highlight novelty in our introduction section

Point 5. What is the main challenge? Why author choose this material? Please highlight in the introduction part.

Response: We are very grateful to Reviewer for reviewing the paper so carefully. We have tried our best to improve the manuscript and have modified some confusing sentences, making them concise and easy to read. We have explain the main challenge and why we choose this material and highlight in the introduction part. We are very grateful to Reviewer for reviewing the paper so carefully.

Point 6. Please check the abbreviations of words throughout the article. All should be consistent.

Response: It is really a detail which should not be neglected as Reviewer suggested and a reference. We have checked the abbreviations of the words throughout the article and make sure they are consistent.

Point 7.  Please don’t use lumpy reference (such as: [1-4,5-9 etc.]). Each reference needs to be properly addressed. Please revise your paper accordingly since same issue occurs on several spots in the paper

Response: It is really a detail which should not be neglected as Reviewer suggested and a reference. In our revised manuscript, we do not use lumpy reference. We have revised that in our revised manuscript. We are very grateful to Reviewer for reviewing the paper so carefully.

Point 8. The main objective of the work must be written on the more clear and more concise way at the end of introduction section. 

Response: It is really a detail which should not be neglected as Reviewer suggested and a reference. We have tried our best to improve the manuscript and have modified some confusing sentences, making them concise and easy to read. We have increase the length of the introduction and explain the novelty of the work. We are very grateful to Reviewer for reviewing the paper so carefully. We have added more references in our introduction section

Point 9.   Please include all chemical/instrumentation brand name and other important specification.

Response: It is really a detail which should not be neglected as Reviewer suggested and a reference. We have revised that in our revised version. We are very grateful to Reviewer for reviewing the paper so carefully.

Point 10.  Please add chemical reagents section and stated all chemical with brand

specifications.

Response: It is really a detail which should not be neglected as Reviewer suggested and a reference. We have revised that in our revised version. We are very grateful to Reviewer for reviewing the paper so carefully.

Point 11.  Please provide space between number and units. Please revise your paper accordingly since some issue occurs on several spots in the paper. 

Response: It is really a detail which should not be neglected as Reviewer suggested and a reference. We have revised that in our revised version. We have provided space between number and units. We are very grateful to Reviewer for reviewing the paper so carefully.

Point 12.   Regarding the replications, authors confirmed that replications of experiment were carried out. However, these results are not shown in the manuscript, how many replicated were carried out by experiment? Results seem to be related to a unique experiment. Please, clarify whether the results of this document are from a single experiment or from an average resulting from replications. If replicated were carried out, the use of average data is required as well as the standard deviation in the results and figures shown throughout the manuscript. In case of showing only one replicate explain why only one is shown and include the standard deviations.

Response: It is really a detail which should not be neglected as Reviewer suggested and a reference. We have added analytical procedures in our revised manuscript. We are very grateful to Reviewer for reviewing the paper so carefully.

Point 13.   Please provide high quality image for figure 2a.

Response: It is really a detail which should not be neglected as Reviewer suggested and a reference. We have provided high quality image of figure 2a in our revised version.

Point14.   Each section of the findings requires thorough discussion; a basic explanation is insufficient; please describe each section in a critical way. 

Response: It is really a detail which should not be neglected as Reviewer suggested and a reference. We have added some explanation in our revised version. We are very grateful to Reviewer for reviewing the paper so carefully.

Point 15.   Please add a comparative profile section to compare your results and prove how it better than previous.

Response: It is really a detail which should not be neglected as Reviewer suggested and a reference. We have added a comparative profile section in our revised version. “Table 1: summary of the anti-mildew property of treated bamboo in different reference”

Point16.  Section 4 should be renamed by Conclusion and Future perspectives. Conclusion section is missing some perspective related to the future research work, quantify main research findings, highlight relevance of the work with respect to the field aspect.

Response: We are very grateful to Reviewer for reviewing the paper so carefully. We have revised that in our revised version. We have added some perspective in our conclusion.

Point 17.   To avoid grammar and linguistic mistakes, Major level English language should be thoroughly checked. Please revise your paper accordingly since several language issue occurs on several spots in the paper.

Response: We are very grateful to Reviewer for reviewing the paper so carefully. We have checked and revised grammar and linguistic mistakes in our revised version.

Point 18.   Reference formatting need carefully revision. All must be consistent in one formate. Please follow the journal guidelines. Reference should not in bold form.

Response: It is really a detail which should not be neglected as Reviewer suggested and a reference. We have revised the reference formatting.

Reviewer 3 Report

Paper is interesting and well written. Changes have to be performed to further improve it:

- chenge the first sentence "Bamboo has become the most promising biomass material in the world" into "Bamboo is one of the most promising biomass materials in the world";

- increase the length of the introduction introducing more literature and at  the end of the introduciton explain also better the novelty of the work;

- at the end of section 2 insert a paragraph named "Analytical procedures", in which you insert also a table with: type of analysis, type of instrument used (model, producer, origin), norms conslulted.

- before the conclusion insert a discussion paragraph where you compare the results of your study with the results of other studies available in literature on bamboo and on other competing materials

Author Response

Paper is interesting and well written. Changes have to be performed to further improve it:

Point 1: chenge the first sentence "Bamboo has become the most promising biomass material in the world" into "Bamboo is one of the most promising biomass materials in the world";

Response: We are very grateful to Reviewer for reviewing the paper so carefully. We have tried our best to improve the manuscript and have modified some confusing sentences, making them concise and easy to read. We have change the first sentence "Bamboo has become the most promising biomass material in the world" into "Bamboo is one of the most promising biomass materials in the world";

Point 2: increase the length of the introduction introducing more literature and at the end of the introduciton explain also better the novelty of the work;

Response: We are very grateful to Reviewer for reviewing the paper so carefully. We have tried our best to improve the manuscript and have modified some confusing sentences, making them concise and easy to read. We have increase the length of the introduction and explain the novelty of the work. We are very grateful to Reviewer for reviewing the paper so carefully.

Point 3: at the end of section 2 insert a paragraph named "Analytical procedures", in which you insert also a table with: type of analysis, type of instrument used (model, producer, origin), norms conslulted.

Response: It is really a detail which should not be neglected as Reviewer suggested and a reference. We have added analytical procedures in our revised manuscript. We are very grateful to Reviewer for reviewing the paper so carefully.

Point 4: before the conclusion insert a discussion paragraph where you compare the results of your study with the results of other studies available in literature on bamboo and on other competing materials

Response: It is really a detail which should not be neglected as Reviewer suggested and a reference. We have added summary of the anti-mildew property of treated bamboo in difference references in our revised version. We are very grateful to Reviewer for reviewing the paper so carefully.

Round 2

Reviewer 1 Report

All the comments have been well addressed. 

Reviewer 2 Report

Accepted

Reviewer 3 Report

accept